# Graph Pre-training and Prompt Learning for Recommendation

## ABSTRACT

GNN-based recommenders have excelled in modeling intricate user-item interactions through multi-hop message passing. However, existing methods often overlook the dynamic nature of evolving user-item interactions, which impedes the adaption to changing user preferences and distribution shifts in newly arriving data. Thus, their scalability and performances in real-world dynamic environments are limited. In this study, we propose GraphPL, a framework that incorporates parameter-efficient and dynamic graph pre-training with prompt learning. This novel combination empowers GNNs to effectively capture both long-term user preferences and short-term behavior dynamics, enabling the delivery of accurate and timely recommendations. Our GraphPL framework addresses the challenge of evolving user preferences by seamlessly integrating a temporal prompt mechanism and a graph-structural prompt learning mechanism into the pre-trained GNN model. The temporal prompt mechanism encodes time information on user-item interaction, allowing the model to naturally capture temporal context, while the graph-structural prompt learning mechanism enables the transfer of pre-trained knowledge to adapt to behavior dynamics without the need for continuous incremental training. We further bring in a dynamic evaluation setting for recommendation to mimic real-world dynamic scenarios and bridge the offline-online gap to a better level. Our extensive experiments including a large-scale industrial deployment showcases the lightweight plug-in scalability of our GraphPL when integrated with various state-of-the-art recommenders, emphasizing the advantages of GraphPL in terms of effectiveness, robustness and efficiency.

### ACM Reference Format:
Anonymous Author(s). 2023. Graph Pre-training and Prompt Learning for Recommendation. In *Proceedings of ACM Conference (Conference'17)*. ACM, New York, NY, USA, 11 pages. https://doi.org/10.1145/nnnnnnn.nnnnnnn

## 1 INTRODUCTION

Recommender systems are integral to numerous Web platforms, assisting users in navigating through the overwhelming amount of information by suggesting relevant items [13, 47]. In recent years, graph neural networks (GNNs) have emerged as powerful tools for modeling user-item interactions in recommendation tasks, enabling effective representation learning on graph-structured data. By treating users and items as nodes and their interactions as edges, GNNs can capture intricate multi-hop relationships between users and items, facilitating the generation of personalized recommendations.

Earlier works [1, 46, 52] on GNN-enhanced recommendation have primarily focused on designing effective message passing mechanisms to capture collaborative relations between users and items. These works aim to leverage the power of GCN to capture

*Conference'17, July 2017, Washington, DC, USA*
2023. ACM ISBN 978-x-xxxx-xxxx-x/YY/MM...$15.00
https://doi.org/10.1145/nnnnnnn.nnnnnnn

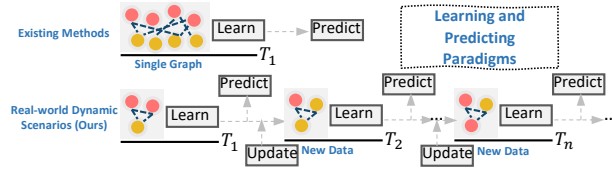

**Figure 1: Our dynamic recommendation setting compared to the vanilla single-graph training in existing methods.**

high-order connectivity in the user-item interaction graph. Subsequent research has further explored simplifying the message passing process [5, 16], reducing the complexity of GNN models [31, 36], and improving the quality of sampling techniques [22]. More recently, researchers have advanced graph-based recommenders by incorporating self-supervised learning (SSL) techniques into GNNs [57]. These methods [2, 49, 56] generally employ the InfoNCE [32] loss to align contrastive views, thereby denoising and improving the robustness of the base LightGCN [16] model.

While these methods have shown impressive performance, they have primarily focused on static scenarios (Figure 1 upper), overlooking the dynamic nature of recommendation, where new user-item interactions continue to evolve over time, often with distribution shifts reflecting changing user interests [50, 51]. In real-world scenarios (Figure 1 lower), it exhibits a dynamic setting, where the model recursively learns from newly arriving data, and predicts for current time. However, existing methods primarily design a model for single-graph training and evaluation, which leads to a degradation in recommendation dynamics and widens the offline-online gap [23]. Additionally, the arrival of new data may exhibit distribution shifts, further complicating the task of making accurate recommendations for graph-based recommendation models without incorporating useful contextual information for the newly arrived data. These challenges significantly limit the scalability of existing models and hinder their ability to adapt to evolving user preferences in a timely manner. This is crucial for providing up-to-date and accurate recommendations in dynamic environments.

To tackle the challenges, we propose a simple yet effective framework named GraphPL, which integrates parameter-efficient and dynamic Graph Pre-Training with Prompt Learning for recommendation. Our proposed method entails pre-training GNNs on extensive historical interaction data, followed by fine-tuning them on more recent target data using time-aware graph prompt learning. In the pre-training phase, the model assimilates knowledge from a substantial amount of historical interactions, effectively capturing long-term user preferences and item relevance. Subsequently, during the fine-tuning phase on the target time periods of data, the model swiftly adapts to evolving user preferences and captures short-term behavior dynamics. This is achieved through a prompt learning schema, facilitating effective knowledge transfer.

To ensure that the pre-trained GNN effectively handles evolving user preferences, our GraphPL framework seamlessly integrates a temporal prompt mechanism alongside a graph-structural prompt

learning mechanism. This integration enables the injection of time-aware context from new data, empowering the model to adapt to changing user preferences. Drawing inspiration from advances in relative positional encoding techniques [34, 39], we meticulously design a dedicated temporal prompt mechanism that aligns with the message aggregation layer of GNNs. Within this prompt mechanism, we encode time information on interaction edges as part of the normalization term for aggregation, all in a parameter-free manner. This innate capability allows the model to naturally incorporate temporal information without the need for additional fine-tuning. By imbuing the pre-trained graph model with temporal awareness, we empower the model to effectively capture vital signals that are more pertinent to the evolving user preferences.

Furthermore, our graph-structural prompt learning mechanism facilitates the seamless transfer of knowledge from the pre-trained model to downstream short-term and recent recommendation tasks. This framework eliminates the requirement for continuous incremental learning of the pre-trained model, instead enabling the transfer of pre-trained knowledge to any future time period to effectively adapt to behavior dynamics. In this mechanism, we incorporate the newly generated interaction edges between the fine-tuning time and the pre-training time as prompt edges. By including these prompt edges, we provide the pretrained model with essential contextual information for the fine-tuning process. Rather than undergoing extensive training, we perform a single non-training forward pass on the prompt edges. This prompts the pretrained model to adapt to the distribution shift of node representations and effectively adjusts its predictions accordingly. It is worthwhile mentioning that our GraphPL is model-agnostic and parameter-efficient, allowing for seamless integration into existing GNN recommenders as a plug-in enhancement. In summary, the main contributions of our work can be summarized as follows:

- We emphasize the criticality of effectively and scalably pre-training and fine-tuning graph-based recommenders for time-evolving user preferences, thus facilitating up-to-date and accurate recommendations in dynamic environments.
- We present GraphPL that effectively handles evolving user preferences by pre-training and fine-tuning GNNs. The proposed prompt learning paradigm facilitates the transfer of valuable and relevant knowledge from the pre-trained model to downstream recommendation tasks in both temporal and structural manners.
- We further introduce a snapshot-based dynamic setting for recommendation evaluation. Compared to the vanilla single-time testing, it brings better approximation to real-world scenarios.
- We conduct extensive experiments on diverse datasets to demonstrate the advantages of GraphPL in terms of robustness, efficiency and performance. To further justify the effectiveness of our framework, we include an online industry deployment with A/B testing on a large-scale online platform.

To ensure result reproducibility, we provide the implementation details and source code of our proposed framework at an anonymous link: https://anonymous.4open.science/r/GraphPL-CC81/.

## 2 PRELIMINARIES

We define the task of pre-training and fine-tuning GNNs for recommendation. We denote the user set as $\mathcal{U}$ and the item set as $\mathcal{I}$.

In the context of collaborative filtering, a typical graph structure, constructed using existing methods [16], can be represented as $\mathcal{G} = (\mathcal{V}, \mathcal{E})$, where $\mathcal{V} = \mathcal{U} \cup \mathcal{I}$ represents the set of all nodes in user-item interaction graph $\mathcal{G}$. The edges in $\mathcal{E}$ correspond to interactions between users and items, with a value of $y_{u,i} = 1$.

In order to provide recommendations at time slot $T_1$, we gather historical user-item interactions to construct a graph $\mathcal{G}_1 = (\mathcal{V}, \mathcal{E}_1)$, where $\mathcal{E}_1$ represents the user-item interactions collected before $T_1$. Existing stationary graph collaborative filtering models typically train the model from scratch using the complete dataset $\mathcal{G}_1$. The objective is to optimize time-specific model parameters $\Theta_1$ by maximizing the likelihood of generating accurate recommendations:

$$\underset{\Theta_1}{\arg\max} \, P_{f_{\Theta_1}}(y_1|\mathcal{G}_1). \tag{1}$$

**Dynamic Learning in Recommender Systems**. In real-world applications, the evaluation of recommenders goes beyond the simplistic static setting commonly used in existing collaborative filtering method [16, 46]. In practice, researchers assess the long-term performance of models by deploying them in a live-update environment, as discussed in [53], where new user-item interactions are continuously generated over time. The model is specifically designed to make ongoing predictions for future user-item interactions based on this evolving data. Formally, the model should have initial weights $\Theta_{n-1}$ corresponding to different time intervals $T_n$, which are then updated through learning on new interactions $\mathcal{G}_n$ to enhance the accuracy of up-to-date predictions.

$$\underset{\Theta_n}{\arg\max} \, P_{f_{\Theta_n}}(y_n|\mathcal{G}_n; \Theta_{n-1}). \tag{2}$$

In this study, we draw inspiration from the work [33] and employ a series of graph snapshots to simulate practical dynamic recommendation scenarios. These graph snapshots are represented as $[\mathcal{G}_1, \mathcal{G}_2, ..., \mathcal{G}_N]$, where we have a total of $N$ snapshots corresponding to different time intervals. Each graph snapshot consists of subsets of users and items from a global set, and the interactions between them evolve over time. Specifically, $\mathcal{G}_n = (\mathcal{V}_n \subset \mathcal{V}, \mathcal{E}_n)$, where $\mathcal{V}_n$ represents the nodes in the snapshot and $\mathcal{E}_n$ denotes the time slot-specific interaction edges. It is important to emphasize that snapshots $\mathcal{G}_n$ are collected within the time period between two consecutive snapshots, namely $[T_{n-1}, T_n]$.

## 3 METHODOLOGY

In this section, we provide the technical details of our proposed GraphPL framework, depicted in Figure 2, which illustrates its architecture. We introduce two crucial components: graph pre-training with a temporal prompt mechanism and a graph-structural prompt-enhanced fine-tuning mechanism. These components are specifically designed to enhance the performance and scalability of GNN-based recommenders in dynamic recommendation scenarios.

### 3.1 Graph Pre-training with Temporal Prompt Mechanism

In practical recommendation scenarios, user-item interaction data continues to accumulate over time. Online platforms like Amazon and TikTok constantly receive new user purchases and video watch logs, respectively, on a daily basis. In such dynamic environments,

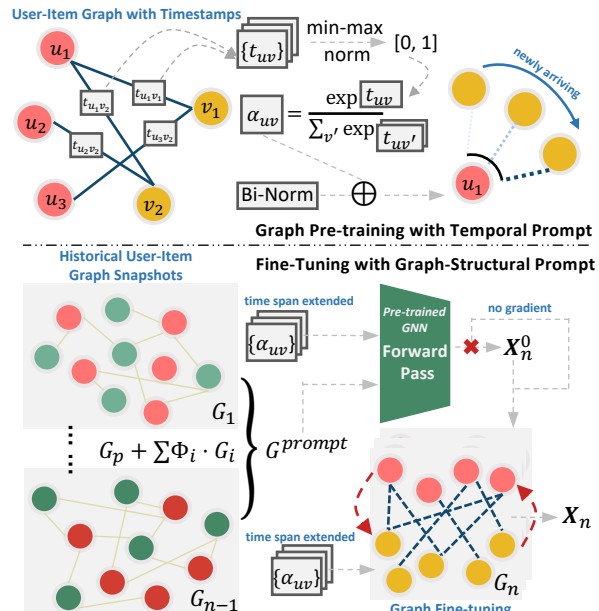

**Figure 2: Overall framework of GraphPL.**

the availability of fresh user-item interactions provides valuable information that can be leveraged to guide pre-trained models in adapting to time-evolving user preferences and providing continuously up-to-date recommendations in dynamic settings.

*3.1.1* **Temporal Prompt Mechanism.** In our framework, we introduce a temporal prompt mechanism to incorporate time-aware contextual information from the latest user preferences and behaviors. This mechanism allows for personalized and timely recommendations by considering the temporal dynamics of user-item interactions. To capture the temporal sequence of user-item interactions, we propose a relative time encoding scheme, which enables us to incorporate temporal information into graph convolutions. By encoding the relative time between interactions, the model can explicitly capture the temporal dependencies and changes in user preferences that are reflected in the newly arrived data.

Our temporal prompt mechanism offers two significant advantages over existing time encoding techniques when it comes to capturing user behavior dependencies across different time slots.

- **Generalization**. Unlike the use of absolute positional embeddings in models like BERT [8], our mechanism takes inspiration from recent advancements in sequence modeling in NLP and leverages a relative positional encoding. Absolute positional embeddings have limited generalization capabilities across continuous time steps, which is problematic in our dynamic recommendation setting. These embeddings are trained on sequences with varying lengths but struggle to handle sequences beyond the trained lengths during fine-tuning and prediction. This leads to a distribution gap between the pretraining phase, where the model learns from fixed-length data, and the fine-tuning and prediction phase, where longer future time steps are encountered.

- **Scalability**. Our temporal prompt design avoids the need to add a fixed-length positional embedding to node representations. Instead, it generates relative temporal-aware weights that can be

seamlessly integrated with message passing. This design allows our pretrained GNN to be easily applied to longer-range graph structures during fine-tuning and testing, greatly improving scalability for dynamic recommendation tasks.

*3.1.2* **Temporal Prompt-enhanced Graph Convolutions.** In order to effectively capture the temporal dynamics of user-item interactions in our model, we have implemented a temporal prompt and incorporated relative time encoding into our graph convolutions. This enables our model to consider the most recent contextual signals from the new data, and adapt to evolving user preferences over time. Within the context of our user-item interaction graph $\mathcal{G}$, the edge attributes consist of Unix timestamps denoted as $t^{\text{unix}}$. These timestamps represent the exact moments when users $u$ interacted with items $v$. To prepare these timestamps for encoding in our model, we convert them into relative time steps by dividing them by a fixed time interval $\tau$. This time interval, which is a hyperparameter, can be defined with a resolution of either hour, day, or week. As a result, for any given edge $e_{u,v}$ in the graph, its corresponding timestep attribute can be computed as follows:

$$t_{u,v} = f_{\text{div}}(t_{u,v}^{\text{unix}}, \boldsymbol{t}^{\text{unix}}) = \lfloor \frac{t_{u,v}^{\text{unix}} - \min(\boldsymbol{t}^{\text{unix}})}{\tau} \rfloor, \quad (3)$$

Here, $\mathcal{T}_{e,v}$ denotes the Unix timestamp assigned to the edge $e_{u,v}$, and the $\lfloor * \rfloor$ notation denotes the floor operation. To avoid the influence of specific numerical scales and ensure uniformity, we normalize these time attributes $\boldsymbol{t} = t_{u,v}|e_{u,v} = 1$ to the range of $[0, 1]$.

$$\boldsymbol{t} = \frac{\boldsymbol{t} - \min(\boldsymbol{t})}{\max(\boldsymbol{t}) - \min(\boldsymbol{t})}. \quad (4)$$

To consider the temporal information among the neighbors during message aggregation in our GNNs, we apply the softmax function to the time attributes $t_{u,v}$ of the first-order neighbors on the graph.

$$\alpha_{u,v} = \frac{e^{t_{u,v}}}{\sum_{v' \in \mathcal{N}_u} e^{t_{u,v'}}}, \quad (5)$$

To enable dynamic time-aware graph neural network (GNN) for recommendation pretraining, we introduce an additional normalization term, $\alpha_{u,v}$, into the message passing step of LightGCN. In this case, $\mathcal{N}_u$ represents the neighbors of node $u$, and $\alpha_{u,v}$ encodes the weight for aggregating information from node $v$ to node $u$.

$$\mathbf{x}_u^{(l)} = \sum_{v \in \mathcal{N}_u} \left( \frac{1}{2\sqrt{|\mathcal{N}_u||\mathcal{N}_v|}} + \frac{\alpha_{u,v}}{2} \right) \mathbf{x}_v^{(l-1)}, \quad (6)$$

We introduce a normalization term and apply mean-pooling to incorporate time-aware normalization into the original bidirectional graph normalization while preserving embedding magnitude.

**Adaptability and Efficiency**. The incorporation of the time-aware normalization term $\alpha_{u,v}$ into the message passing of LightGCN enhances the GNN's adaptability to evolving user-item interactions over time. By giving more weight to interactions that are closer in time during neighbor aggregation, the model becomes more attentive to the dynamic nature of user-item interactions and assigns higher importance to recent interactions. This alignment with the objective of recommendation tasks ensures that the model captures timely and relevant user preferences, leading to more accurate and personalized recommendations. Importantly, our time-aware regularization approach does not introduce additional embedding

encoding. Instead, it dynamically generates graph regularization terms based on the relative time order, making it a lightweight and efficient solution. This design allows the model to handle varying absolute time lengths effectively, showcasing excellent generalization capabilities and requiring minimal computational overhead.

## 3.2 Fine-Tuning with Graph-Structural Prompt Mechanism

In this section, we will discuss how we effectively transfer knowledge from a pre-trained Graph Neural Network (GNN) model for fine-tuning with future user-item interactions. To begin the fine-tuning process at the target time $T_n$, which occurs after the pre-training time $T_p$, the intuitive way is to update the model parameters incrementally by simply fine-tuning. That is, we iteratively provide the model with data from the updated time intervals to fine-tune the node representations that were previously updated in the preceding time intervals. Therefore, the initial embeddings for fine-tuning at $T_n$ are derived from the forward pass after the last fine-tuning step.

$$\mathbf{X}_n^0 = \text{forward}(\mathbf{X}_{n-1}; \mathcal{G}_{n-1}), \quad (7)$$

where forward($\ast$) represents the complete forward pass of the model, utilizing the last fine-tuned embeddings $\mathbf{X}_{n-1}$ and the graph structure $\mathcal{G}_{n-1}$. This method has the advantage of directly capturing users' continuous interest changes within a specific time span. However, the incremental fine-tuning mechanism has two significant drawbacks. First, iteratively updating model parameters based on small-range interactions may lead the model to converge to a local optimum specific to that time period, limiting the potential for continuous fine-tuning on the updated representations in the future. Secondly, persistently updating the parameters of the pre-trained model can result in a significant computational burden.

### 3.2.1 Graph-Structural Prompt Mechanism.
In our approach, we address the mentioned issues by leveraging the interaction edges between the pre-training time $T_p$ and the current time $T_n$ as prompt edges. This allows the pretrained model to directly fine-tune on future time periods without the need for iterative updates. Inspired by discrete prompt tuning in large language models [35, 37], we treat the edges of the graph during a specific time period as discrete prompts that guide the propagation of pretrained embeddings. This captures the representation shift between the pre-training and fine-tuning time points and provides better temporal-aware initial embeddings for fine-tuning. To generate prompt structures, we concatenate the pre-training graph structure with the sampled future edges between the pre-training and current fine-tuning time. This combination enables the model to capture the temporal dynamics and improve the effectiveness of fine-tuning:

$$\mathcal{G}^{\text{prompt}} = \mathcal{G}_p \oplus \sum_{i=1}^{n} \Phi_i \odot \mathcal{G}_i; \ \Phi_i = \begin{cases} 1 - (i-1)\phi, & \phi > 0 \\ 1 + (n-i)\phi, & \phi < 0 \end{cases}, \quad (8)$$

where "$\oplus$" denotes graph concatenation and "$\odot$" denotes graph sampling. Here, a hyper-parameter $\phi$ is introduced as the sampling decay for prompt structures, where a positive $\phi$ suggests that we include more early structures and less recent ones, and vice versa. After generating the prompt structures, we proceed with a forward pass using the pretrained embeddings $\mathbf{X}_p$ on the prompt graph to generate embeddings for fine-tuning. To mitigate the overfitting

effect and improve generalization for more robust fine-tuning in our GraphPL framework, we introduce a random gating [3] mechanism that slightly perturbs the pre-trained embeddings.

$$\widetilde{\mathbf{X}}_p = \mathbf{X}_p \odot \text{sigmoid}(\widetilde{\mathbf{W}}\mathbf{X}_p + \widetilde{\mathbf{b}}), \quad (9)$$

$$\mathbf{X}_n^0 = \text{forward}(\widetilde{\mathbf{X}}_p; \mathcal{G}^{\text{prompt}}), \quad (10)$$

The non-learnable random gating weights, $\widetilde{\mathbf{W}} \in \mathbb{R}^{d \times d}$ and $\widetilde{\mathbf{b}} \in \mathbb{R}^d$, are generated from a Gaussian distribution. It's important to note that the relative time encoding also plays a vital role in facilitating the model's ability to sense relative temporal connections during the prompt propagation process. By propagating the embeddings learned from extensive pretraining over a large time period on the prompt edges, which include interactions from subsequent time periods, we achieve two objectives. Firstly, we enable the obtained embeddings to maintain stable user interests. Secondly, we swiftly capture changes in user interests within the subsequent time span. By refraining from directly training the embeddings on the short-term graph, we mitigate the risk of the model parameters becoming trapped in local optima. This approach grants us greater flexibility for subsequent fine-tuning and enables the model to more effectively adapt to users' evolving interests over time.

### 3.2.2 Prompt Learning with Adaptive Gating Mechanism.
To address the distribution shift in node representations between the time-aware graph snapshots $\mathcal{G}_{n-1}$ and $\mathcal{G}_n$, we introduce a learnable gating mechanism that adaptively transforms the input embeddings $\mathbf{X}_n^0$. This gating mechanism allows for modeling the changes in user/item representations over time, effectively preserving the informative signals necessary for making accurate future recommendations. We employ gradient truncation on $\mathbf{X}_n^0$ to prevent direct optimization of the large-scale pre-trained model. Instead, we fine-tune $\mathbf{X}_n^0$ using newly interaction structual contexts $\mathcal{G}_n$ to improve the accuracy of predictions at the target time interval $T_n$.

To prevent direct optimization of the large-scale pre-trained model, we employ gradient truncation on $\mathbf{X}_n^0$. Instead, we fine-tune $\mathbf{X}_n^0$ using the newly observed interaction structural contexts $\mathcal{G}_n$, which helps improve the prediction accuracy specifically for the target time interval $T_n$. By combining the gating mechanism and gradient truncation, we can adaptively update the embeddings while mitigating the impact of distribution shifts, ensuring the model's ability to capture temporal dynamics with good adaptiation.

$$\widetilde{\mathbf{X}}_n^0 = \mathbf{X}_n^0 \odot \text{sigmoid}(\mathbf{W}_l \mathbf{X}_n^0 + \mathbf{b}_l), \quad (11)$$

$$\mathbf{X}_n = \text{forward}(\widetilde{\mathbf{X}}_n^0; \mathcal{G}_n), \quad (12)$$

At this stage, we have derived the user and item representations $\mathbf{X}_n$ for making predictions starting from time $T_n$. In this process, the learnable weights $\mathbf{W}_l$ and $\mathbf{b}_l$, which have the same size as the random gating, are introduced. To estimate the probability of user $u$ interacting with item $i$, we calculate the dot product between the user and item representations $\mathbf{x}_n^u$, $\mathbf{x}_n^i$, denoted as $\hat{y}_{u,i} = \mathbf{x}_n^{u\mathsf{T}} \cdot \mathbf{x}_n^i$.

## 3.3 Model Learning and Discussion

### 3.3.1 Optimized Objective.
In both the pre-training and fine-tuning stages, we define our training objectives based on optimizing the BPR loss. The BPR loss ensures that the predicted score for an observed interaction is higher than that of its unobserved negative

samples. This loss function is commonly used in recommendation systems to model the preference ranking between items for individual users. By optimizing this loss, we aim to improve the model's ability to accurately rank and predict user-item interactions.

$$\mathcal{L} = - \sum_{(u,i,j) \in D} \log \sigma(\hat{y}_{ui} - \hat{y}_{uj}), \quad (13)$$

In our training strategy, we utilize a dataset $D$ that includes negative items $j$ sampled at each training mini-batch. Our approach follows a two-stage process. In the first stage, we pre-train a GNN-based recommender on a large-time-scale graph until convergence. This involves training the model on a comprehensive set of historical data, allowing it to learn long-term patterns and user preferences. In the second stage, we fine-tune the pre-trained model on small-time-scale graph snapshots that include interactions from a more recent time period. This fine-tuning process helps the model adapt and capture short-term changes in user interests and item dynamics.

*3.3.2* **Interplotive Parameter Update.** To ensure that the model parameters are learned in synchronization with the evolving user and item representations, it is important to update the pre-trained node embeddings over time steps. Inspired by the investigation in [33, 53], we propose an interpolative approach for updating the pre-trained user and item embeddings. Specifically, to estimate the best initial state for training at the next time step $T_n$, we combine the pre-trained embeddings with the embeddings learned within a sliding window $[T_{n-\omega}, T_{n-1}]$ using interpolation. This allows us to leverage both the long-term knowledge captured during pre-training and the recent changes observed within the sliding window, enabling the model to effectively adapt to the evolving dynamics of user-item interactions:

$$\mathbf{X}_n^{\text{init}} = \text{mean}(\mathbf{X}_p, \sum_{i=1}^{\omega} \frac{i \cdot \mathbf{X}_{n-i}}{\sum_{k=1}^{\omega} k}) \quad (14)$$

$\mathbf{X}$ represents the model parameters, which correspond to the user and item embeddings. The left term of the equation calculates a weighted normalization of the weights $[\mathbf{X}_{n-1}, ..., \mathbf{X}_{n-\omega}]$, where the more recent fine-tuned representations are given less weight. This weighting helps to mitigate the local optima effect, where the model may get stuck in suboptimal solutions based on recent but noisy information. The hyperparameter $\omega$ controls the size of the sliding window, which determines the number of previous time steps considered for fine-tuning. As $\omega$ becomes smaller, the model updates its evolved representations more frequently, allowing it to capture recent changes. However, this can increase the risk of getting trapped in local optima due to the limited historical information considered. On the other hand, if $\omega$ is larger, the model can incorporate longer-term information, but may have reduced sensitivity to recent fine-tuned weights.

## 4 EVALUATION

In this section, we compare our proposed GraphPL with state-of-the-art methods across diverse research lines and settings, with the aim of addressing the research questions shown below.

- **RQ1**: Can GraphPL outperform state-of-the-art time-aware graph learning models and pre-trained GNNs in making dynamic recommendations across different time slots?

- **RQ2**: How does GraphPL perform when integrated as a model-agnostic plug-in component with state-of-the-art recommenders?
- **RQ3**: Can GraphPL perform on par or even outperform the vanilla full-data training paradigm?
- **RQ4**: How does the performance of GraphPL change under different ablation settings of key components and hyper-parameters?
- **RQ5**: How effective is GraphPL in tackling the cold-start issue?
- **RQ6**: How does the potential scalability of GraphPL facilitate efficient model convergence with our prompt learning paradigm?
- **RQ7**: Can GraphPL effectively empower real-world recommendation systems when deployed in industrial applications?

### 4.1 Experimental Settings

*4.1.1* **Datasets.** We adopt three public datasets covering diverse real-world scenarios of dynamic recommendation. **Taobao** records implicit feedback from Taobao.com, a Chinese e-commerce platform during 10 days. **Koubei** dataset, provided for the IJCAI'16 contest, records 9 weeks of user interactions with nearby stores on Koubei in Alipay. **Amazon** dataset consists of a 13-week's collection of product reviews sourced from Amazon. The details can be referred in Table 3.

*4.1.2* **Baseline Models.** We include the recent dynamic graph neural networks and graph prompt approaches as our baselines. Specifically, three most relevant research lines are included for comparison: **Dynamic Recommendation Methods**: DGCN [26], which formulates a dynamic learning task on the single graph. **Graph Prompt Methods**: GraphPrompt [29] and GPF [11]. **Dynamic Graph Neural Networks**: EvolveGCN-O, EvolveGCN-H[33] and ROLAND [53]. For detailed descriptions regarding the baseline methods, please refer to Appendix A.1.1.

*4.1.3* **Integration with GNN Recommenders.** To highlight its versatility, GraphPL serves as a general architecture that can be seamlessly integrated as a plug-in component with any GNN-based recommender. In our evaluation, we implement GraphPL using the LightGCN [16] model, renowned for its simplicity and efficiency. Furthermore, we extend the applicability of GraphPL by integrating it with SOTA recommenders that incorporate self-supervised learning designs, such as SGL [49], MixGCF [22], and SimGCL [56], providing empirical evidence of the GraphPL's effectiveness in enhancing them for dynamic and adaptable recommendations.

*4.1.4* **Evaluation Protocols.** Our evaluation settings encompass graph snapshots of multiple time intervals (e.g., day, week) to further simulate the practical challenges of learning dynamics. To learn from snapshots, we fine-tune all the models using a 2-size sliding window approach to continuously learn from the current snapshot and make predictions for the next. For the proposed P-L paradigm, we take a large portion of the data to pre-train, and use the remaining snapshots for fine-tuning and testing as shown in Table 3. To ensure fair comparison free of data inequality, the baselines also follow the same proposed paradigm. For example, for dynamic GNNs, the very initial weights for tuning are also trained over the pre-training time span. We report metrics averaged over all target future temporal snapshots as following the common setting in [33, 53]. We report Recall@k and nDCG@k at k=20, following the common all-rank settings in previous works [16, 17, 49].

**Table 1: When compared to various baselines utilizing different backbone architectures, GraphPL consistently exhibits strong overall performance across different types of datasets. The script ∗ denotes the statistically significant results compared to the second best at $p < 0.01$ level.**

| Method | Taobao | | Koubei | | Amazon | |
|---|---|---|---|---|---|---|
| | Recall | nDCG | Recall | nDCG | Recall | nDCG |
| DGCN | 0.0229 | 0.0228 | 0.0353 | 0.0255 | 0.0158 | 0.0084 |
| **LightGCN+** | | | | | | |
| GraphPrompt | 0.0199 | 0.0195 | 0.0342 | 0.0249 | 0.0154 | 0.0075 |
| GPF | 0.0223 | 0.0220 | 0.0348 | 0.0251 | 0.0174 | 0.0088 |
| EvolveGCN-H | 0.0224 | 0.0221 | 0.0315 | 0.0231 | 0.0138 | 0.0066 |
| EvolveGCN-O | 0.0236 | 0.0232 | 0.0334 | 0.0242 | 0.0157 | 0.0084 |
| ROLAND | 0.0226 | 0.0226 | 0.0301 | 0.0223 | 0.0150 | 0.0069 |
| **GraphPL** | **0.0251**∗ | **0.0245**∗ | **0.0362**∗ | **0.0265**∗ | **0.0191**∗ | **0.0094**∗ |
| **SGL+** | | | | | | |
| GraphPrompt | 0.0223 | 0.0220 | 0.0355 | 0.0261 | 0.0161 | 0.0079 |
| GPF | 0.0229 | 0.0226 | 0.0363 | 0.0266 | 0.0187 | 0.0096 |
| EvolveGCN-H | 0.0235 | 0.0232 | 0.0358 | 0.0263 | 0.0137 | 0.0066 |
| EvolveGCN-O | 0.0242 | 0.0238 | 0.0365 | 0.0268 | 0.0173 | 0.0090 |
| ROLAND | 0.0222 | 0.0222 | 0.0340 | 0.0251 | 0.0161 | 0.0078 |
| **GraphPL** | **0.0268**∗ | **0.0264**∗ | **0.0371**∗ | **0.0277**∗ | **0.0221**∗ | **0.0114**∗ |
| **MixGCF+** | | | | | | |
| GraphPrompt | 0.0248 | 0.0245 | 0.0377 | 0.0276 | 0.0180 | 0.0089 |
| GPF | 0.0251 | 0.0247 | 0.0380 | 0.0278 | 0.0182 | 0.0092 |
| EvolveGCN-H | 0.0240 | 0.0237 | 0.0354 | 0.0262 | 0.0129 | 0.0061 |
| EvolveGCN-O | 0.0271 | 0.0267 | 0.0375 | 0.0276 | 0.0171 | 0.0085 |
| ROLAND | 0.0232 | 0.0230 | 0.0349 | 0.0260 | 0.0152 | 0.0072 |
| **GraphPL** | **0.0280**∗ | **0.0273**∗ | **0.0393**∗ | **0.0291**∗ | **0.0216**∗ | **0.0109**∗ |
| **SimGCL+** | | | | | | |
| GraphPrompt | 0.0239 | 0.0224 | 0.0348 | 0.0258 | 0.0139 | 0.0069 |
| GPF | 0.0237 | 0.0220 | 0.0357 | 0.0264 | 0.0182 | 0.0094 |
| EvolveGCN-H | 0.0241 | 0.0238 | 0.0356 | 0.0265 | 0.0134 | 0.0067 |
| EvolveGCN-O | 0.0241 | 0.0238 | 0.0351 | 0.0258 | 0.0168 | 0.0088 |
| ROLAND | 0.0228 | 0.0228 | 0.0333 | 0.0246 | 0.0151 | 0.0075 |
| **GraphPL** | **0.0280**∗ | **0.0276**∗ | **0.0368**∗ | **0.0276**∗ | **0.0205**∗ | **0.0108**∗ |

## 4.2 Performance Comparison (RQ1–RQ3)

*4.2.1 Comparison with Baselines.* We present the performance of our GraphPL method as well as other alternative solutions, including graph prompt methods and dynamic GNNs, with LightGCN as the base model, as summarized in Table 1. Analyzing the results, we have made the following key observations:

- Our GraphPL consistently outperforms graph prompt and dynamic graph learning methods, demonstrating the superiority of our pre-training and prompt learning design. Specifically, we observe average improvements of 6.0%, 3.2%, and 8.3% in Recall and nDCG across the three datasets. We attribute these advantages to two key factors: 1) Our temporal prompt mechanism empowers GraphPL to capture dynamically evolving user-item interactions throughout both the pre-training and fine-tuning stages. 2) The graph-structural prompt design facilitates effective knowledge transfer from the large-scale pre-trained model, addressing distribution shift issues between temporal snapshots.

- The absence of a consistent winner among the baseline methods indicates the challenging nature of the dynamic recommendation task. Despite EvolveGCN's impressive performance on the Taobao dataset, it does not exhibit a distinct advantage over other models on the remaining datasets. This discrepancy may

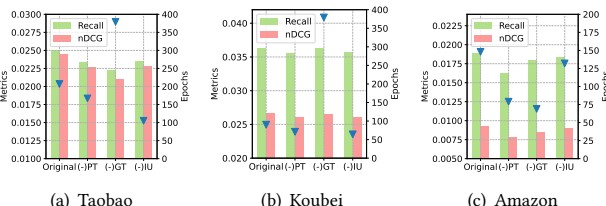

| (a) Taobao | (b) Koubei | (c) Amazon |
|---|---|---|

**Figure 3: Key component ablation study for fine-tuning stage. Y-axis denotes performance metrics on the left and epochs (displayed as ▽) for convergence on the right.**

be due to potential overfitting issues related to short-term user patterns, resulting from the use of complex neural architectures to encode user-item interactions in each new time period. In contrast, our GraphPL utilizes lightweight prompt mechanisms that effectively capture both long-term user interests and incorporate new preference context from recently observed behavior data.

- Despite being regarded as the meticulously-designed dynamic GNN, ROLAND does not demonstrate superior performance. This limitation may be attributed to its intricate model parameter update schemes, which introduce larger perturbations to embeddings. Consequently, the representation learning for users and items is disrupted, rendering it less effective in capturing the time-evolving user preferences in recommendation tasks.

*4.2.2 Integration with SOTA Methods.* Additionally, we assess the adaptability of GraphPL across different backbone recommenders, namely MixGCF, SGL, and SimGCL. We re-implement all methods on these base models using the same evaluation settings. The evaluation results, averaged across multiple time slots, are presented in Table 1. Our observations are summarized as follows:

- GraphPL continues to demonstrate superior performance when integrated with state-of-the-art recommenders. This highlights the remarkable adaptability of our approach, enhancing the performance of different state-of-the-art models in diverse scenarios. Baseline methods exhibit diverse performance rankings across different base recommenders and datasets. No single method consistently outperforms others across all three datasets. On Taobao, EvolveGCN-O is generally on par as the second-best method. However, on Amazon, GPF consistently outperforms other baselines with all three base recommenders. This sensitivity to data characteristics and base models hinders the baseline methods from yielding stable and significant results.

- In general, better representation learning capabilities in the base model lead to higher performance. When comparing GraphPL's performance in the SGL+Taobao and SimGCL+Taobao settings, there is a 4.5% improvement. This indicates that our approach effectively benefits from the enhanced representation provided by the base model and performs exceptionally well in dynamic scenarios. However, methods like EvolveGCN-O do not show improved performance as the base model's representation capabilities increase. This suggests that their designs may lack strong generalization ability and could even yield negative outcomes.

*4.2.3 Comparison with Full-Data Training.* We place the results and discussion in Appendix A.2.2 due to space limitation.

## 4.3 Ablation Study (RQ4)

*4.3.1* **Key Components in Fine-Tuning.** We conduct a comprehensive ablation study to examine the effectiveness of the key components in the design of GraphPL, both in the pre-training and fine-tuning stages. To facilitate comparisons with the original design, we create three variants of GraphPL, with each variant removing one key component. Specifically, these variants are:

- (-)PT: We disable the **P**rompt **T**uning module, which utilizes prompt edges derived from historical interactions. Instead, we directly fine-tune the pretrained weights using new edges.
- (-)GT: We suppress the adaptive **Ga**ting mechanism during the fine-tuning, which endows dynamic knowledge transformation.
- (-)IU: We exclude the **I**nterplotive **U**pdate module and keep the pretrained weights unchanged for each fine-tuning step.

Based on the results in Figure 3, we make the following observations: 1) All three key components contribute positively to our GraphPL design. Removing any component leads to a significant decrease in recommendation accuracy and, in some cases, longer convergence epochs. This demonstrates the effectiveness of these components. 2) The structural prompt and interplotive update mechanisms enhance accuracy and mitigate local optima effects in dynamic learning. Disabling these components allows faster convergence but results in significantly worse accuracy. 3) The adaptive gating mechanism accelerates model convergence by facilitating better gradient discovery, leading to improved accuracy. Removing the gating mechanism results in substantially longer convergence epochs and worse accuracy, suggesting its importance in addressing the distribution gap between the fine-tuning on snapshots.

*4.3.2* **Effect of Pre-trained Model.** We examine the impact of different pretrained model designs on downstream fine-tuning performance by comparing four variants. These variants investigate the effects of relative time encoding and representation power of the pretrained models. The original model, "LGN(+)TE," is trained on LightGCN with time encoding (TE). "LGN(-)TE" removes the time encoding design during pretraining. Additionally, "MixGCF(+)TE" and "SimGCL(+)TE" utilize stronger models for pretraining while keeping the fine-tuning model as LightGCN. Figure 4 presents the performance comparison of both pretraining and fine-tuning stages.

- The temporal prompt mechanism significantly accelerates convergence and improves prediction accuracy in both pretraining and fine-tuning stages. It effectively guides the model to leverage important temporal information during message passing.
- Stronger pretrained models yield better performance, aligning with findings in pre-trained language [8] and vision models [9]. This demonstrates the scalability and adaptability of our framework, enabling powerful pretrained models to achieve superior performance. It highlights the potential of leveraging large pretrained models in recommendation tasks, fine-tuning powerful embeddings with lightweight models for downstream benefits.

## 4.4 Learning Impact Analysis (RQ5 & RQ6)

*4.4.1* **Fine-tuned v.s. Untuned (Cold-start) Nodes.** This section analyzes how the fine-tuning design benefits the learning of node representations for recommendation. We categorize users into two groups based on whether they undergo fine-tuning during each time step. The untuned users represent cold-start users in that period.

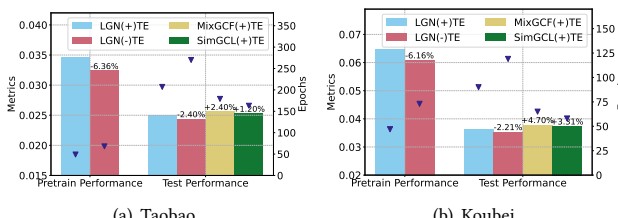

| (a) Taobao | (b) Koubei |
|---|---|

**Figure 4: Ablation study for pretrained models.**

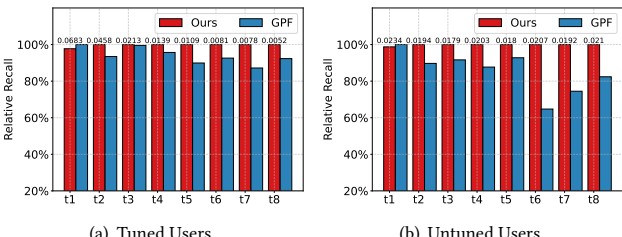

| (a) Tuned Users | (b) Untuned Users |
|---|---|

**Figure 5: Evaluation performance for tuned and untuned users on Amazon compared with the best baseline, GPF.**

On the Amazon data, we evaluate the two user groups separately at each time step, visualizing the results in Figure 5.

- GraphPL effectively learns refined representations for both tuned and untuned (cold-start) users. It achieves dominant performance for both groups in most cases. This advantage is attributed to the structural prompt design, where previous interactions provide informative knowledge for improving representation learning.
- GraphPL exhibits greater performance enhancement from long-term tuning and prediction compared to the baseline. While the baseline initially outperform GraphPL in the first snapshot, GraphPL consistently outperforms the strongest baseline from $T_2$ to $T_8$. This demonstrates the superiority of our GraphPL design in mitigating local optima effects and achieving better long-term gains in dynamic learning scenarios.

*4.4.2* **Efficiency in Learning.** This section focuses on the learning efficiency of our GraphPL design. GraphPL is a parameter-efficient method that minimizes the number of learnable weights for pretraining and fine-tuning, in contrast to incremental training methods. This leads to minimal additional training cost in terms of time and computation. To demonstrate the efficiency gains and faster convergence of our design, we compare the training curves of GraphPL with the second baseline methods (EvolveGCN-O and GPF) on the Taobao and Koubei datasets, as shown in Figure 6. The results clearly indicate that GraphPL achieves significantly better performance while requiring fewer learning epochs. For instance, GraphPL converges after four fine-tuning stages, consuming approximately half the epochs and training time compared to the baselines. These findings highlight the substantial learning efficiency improvements offered by GraphPL.

## 4.5 Online Deployment and A/B Test (RQ7)

We deploy GraphPL on a large-scale online content consumption platforms (specific name withheld due to anonymous requirements) with millions of users, to evaluate its effectiveness in personalized content recommendation. We integrate GraphPL with the main

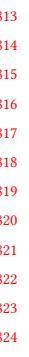
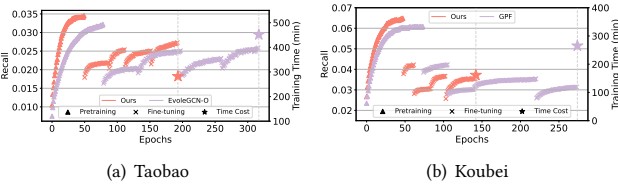

(a) Taobao

(b) Koubei

**Figure 6: The training curves for GraphPL and the baselines on the Taobao and Koubei datasets are shown. The scatters indicate the evolving performances across different stages (pre-training and fine-tuning) with respect to training epochs. The star marker represents the final convergence point, and the right y-axis represents the overall time consumption.**

**Table 2: Online A/B test results spanning 5 days. HPC: highly-personalized content. CC: click count. VCC: video click count.**

| Model | CTR | HPC CTR | Avg. CC | Avg. VCC |
|---|---|---|---|---|
| Online Model | 10.61% | 13.42% | 0.6716 | 0.0188 |
| GraphPL | 10.78% | 13.89% | 0.6831 | 0.0194 |
| # Improve | 1.53%± 0.68% | 3.45%± 0.64% | 1.71%± 0.84% | 3.28%± 1.76% |

CTR prediction model, utilizing user embeddings trained with unsupervised deep graph infomax (DGI) as pretrained weights. Prompt edges are created using historical item-to-user interactions, and the pretrained user embeddings are fine-tuned with a 1-layer GNN to derive item representations for training the main model. The pretrained embeddings are updated synchronously with the main model at a ten-minute granularity. In the online A/B test, we allocate an equal user engagement of around 2 million users to GraphPL and the online model separately. We evaluate the performance on 4 metrics related to CTR and click count over a 5-day period, as shown in Table 2. The results demonstrate that GraphPL significantly improves the real-world recommender system by effectively modeling evolving user and item representations and leveraging deep user interests through pretraining and fine-tuning. Notably, GraphPL is easy to deploy with minimal effort, making it a cost-effective solution for enhancing online recommender systems.

## 5 RELATED WORKS

**GNNs for Recommendation.** GNNs have gained prominence in recommenders for extracting multi-hop collaborative signals [12]. NGCF [46] and PinSage [52] are popular GNN models that refine user and item embeddings recursively using message-passing. GCCF [6] introduces a residual structure, while LightGCN [16] simplifies the architecture by removing non-linear transformations. Researchers have also explored extending GNNs to model complex collaborative relationships, using techniques such as hypergraph learning [27, 55] and intent disentanglement [45, 48]. Self-supervised learning (SSL) has recently been applied to address sparsity issues in graph-based recommenders, with methods like contrastive learning on user-item graphs [25, 38].

**Pre-training and Fine-tuning on GNNs.** Inspired by successful pre-training and fine-tuning in NLP, researchers have explored empowering GNNs with similar techniques. Strategies like contrastive learning and infomax-based pre-training have been developed for better representation learning [20, 41, 44, 54]. Pre-training methods, such as link prediction and feature generation, and prompt-based

fine-tuning have also been proposed [11, 18, 21, 29, 42, 43]. However, these approaches have not fully addressed the challenges of pre-training and fine-tuning in dynamic graph learning, leading to suboptimal performance in temporal-transfer tasks. In contrast, our proposed GraphPL excels in temporally dynamic graph learning under the pre-training and fine-tuning paradigm.

For recommendation tasks, [15, 28] propose pre-training models specifically for user and item modeling. In [15], a GNN is pre-trained as a pretext task to simulate the cold-start scenario. In [28], a side-information-based pre-training scheme is designed. However, these methods primarily focus on stationary recommendation scenarios and overlook the time-evolving nature of user preferences. As a result, their generalization as time-aware recommenders is limited.

**Dynamic Graph Learning.** Learning on temporally evolving graphs is an emerging research trend. Existing approaches, including EvolveGCN [33], Dyngraph2vec [14], DGNN [30], ROLAND [53], and WinGNN [59], employ various techniques such as RNNs, dense layers, and recurrent layers to capture graph dynamics. However, these methods lack a pre-training and fine-tuning framework for dynamic graph learning, and may introduce noisy perturbations to user and item representations. DGCN [26] incorporates dynamics in graph-based recommendation learning, but it does not explicitly consider a dynamic graph setting with snapshots and is limited to evaluation within a single graph.

**Sequential Recommendation.** Sequential recommendation, also known as next-item recommendation, is another research area that focuses on temporal-aware recommendation settings. Representative works in this field include i) attention-based methods: SASRec [24], BERT4Rec [40], STOSA [10]; ii) GNN-based recommenders: SURGE [4] amd Retagnn [19]; iii) SSL-enhanced models: S3-rec [58] and ICL [7]. While our approach also considers temporal dynamics, it differs significantly from next-item recommendation methods. Firstly, sequential recommenders typically employ auto-regressive encoders, which limit their capability to accurately predict the next item and are not directly comparable to graph-based methods that can recall top-K items. Secondly, existing sequential methods do not explicitly consider time in terms of daily or weekly intervals. Instead, they retrieve a fixed-length historical sequence.

## 6 CONCLUSION

This study proposes a novel framework, GraphPL, that integrates dynamic graph pre-training with prompt learning to improve the adaptation and scalability of time-aware recommender systems. By employing a temporal prompt mechanism, our framework enables the transfer of valuable knowledge from the pre-trained model to downstream recommendation tasks on newly arrived data. In addition, the inclusion of graph-structured prompt learning with adaptive gating mechanism, allowing for the incorporation of crucial contextual information, facilitating fine-tuning and adaptation to changing behavior dynamics. Through a comprehensive set of experiments on diverse real-world datasets, we demonstrate that GraphPL outperforms state-of-the-art baselines in making dynamic recommendations across different time slots. Our future work lies in investigating the interpretability of the prompt graph edges in GraphPL, which can provide insights into the contextual information that is used to fine-tune the pre-trained GNNs.

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

# A APPENDIX

## A.1 Evaluation Details

*A.1.1* **Baseline Models.** Here are detailed descriptions of the baseline models used as competitors in our model evaluation:

**Dynamic Recommendation Methods.** We include DGCN [26] which also studies dynamic learning in collaborative filtering. It categorizes edges as past and current ones, and designs a new GNN framework that makes information flow from past edges to current. However, it does not include an explicit dynamic setting with snapshots, and focuses learning on a single graph.

**Graph Prompt Methods.** This line aims to unify the pre-training and downstream tasks using a common template while leveraging prompts for task-specific knowledge retrieval.

- GraphPrompt [29]. It introduces an approach to pretraining and prompting in the context of graphs. It utilizes a learnable prompt to guide downstream tasks, enabling them to access relevant knowledge from pretrained models using a shared template.
- GPF [11]. This method introduces prompts within the feature space of the graph, thereby establishing a general approach for tuning prompts in any pre-trained graph neural networks.

**Dynamic Graph Neural Networks.** They focus on addressing dynamic graphs by updating previously learned embeddings in a time-aware manner, and handling graph dynamics. As competitors for comparison, we include EvolveGCN [33] and ROLAND [53].

- EvolveGCN [33]. This method addresses the dynamism of graph sequences by utilizing an RNN to adapt the parameters of the Graph Convolutional Network (GCN) over time. The GCN parameters can be either hidden states (referred to as the -H variant) or inputs of a recurrent architecture (referred to as the -O variant).
- ROLAND [53]. This state-of-the-art dynamic graph learning baseline utilizes a meta-learning approach to update previously learned embeddings for re-initialization. These updated embeddings are then fused with layer-wise hidden states of GNN.

**Table 3: Statistics of the experimental datasets.**

| Statistics | Taobao | Koubei | Amazon |
|---|---|---|---|
| # Users | 117,450 | 119,962 | 131,707 |
| # Items | 86,718 | 101,404 | 107,028 |
| # Interactions | 8,795,404 | 3,986,609 | 876,237 |
| # Density | 8.6e-4 | 3.3e-4 | 6.2e-5 |
| Temporal Segmentation | | | |
| # Pre-training Span | 5 days | 4 weeks | 4 weeks |
| # Tuning-Predicting Span | 5 days | 5 weeks | 9 weeks |
| # Snapshot Granularity | daily | weekly | weekly |

## A.2 Additional Experiments

*A.2.1* **Hyper-parameter Sensitivity.** Here we study how sensitive GraphPL is towards hyper-parameter settings change. On the three datasets, we include all the hyper-parameters of our GraphPL design, which are time interval $\tau$ for the temporal prompt, updating window $\omega$ for interplotive update, and sampling decay $\phi$ for the structural prompt. The search space varying from datasets are:

- $\tau$: Taobao-[0.5, 1, 4, 12]; Koubei and Amazon-[24, 48, 72, 96]
- $\omega$: Taobao and Koubei-[1, 2, 3]; Amazon-[2, 4, 6]
- $\phi$: [0.05, 0.1, −0.05, −0.1]

We present the results in Figure 8 and summarize our findings as: 1) Generally, GraphPL is more sensitive to these hyper-parameters on Amazon than on Taobao and Koubei. This may due to the longer fine-tuning and predicting span on Amazon (9 weeks). This suggests that the hyper-parameters in GraphPL have larger effect on long-term performances. To obtain better long-term performances, the hyper-parameters should be chosen more carefully. 2) GraphPL is more sensitive to updating window $\omega$ than other HPs. A shorter window size may be insufficient to include useful updates of user and item representations, while a longer one may risk bringing more noise that is relevant to current representation learning. 3) Overall, small adjustments to these hyper-parameters do not lead to significant performance degradation, indicating the robustness of our model to hyper-parameter variations. We recommend selecting suitable hyper-parameters based on the data characteristics.

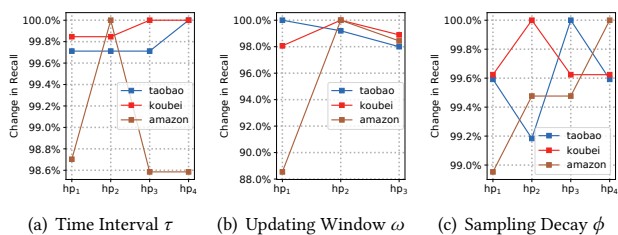

(a) Time Interval $\tau$  (b) Updating Window $\omega$  (c) Sampling Decay $\phi$

**Figure 8: Performance change w.r.t. key hyperparameters.**

*A.2.2* **Comparsion with Full-Data Training.** In this section, we investigate whether GraphPL can perform on par or even outperforms the vanilla full-data training method ("FULL"). On all three datasets, we analyze the performances of GraphPL and FULL on every snapshot in terms of Recall and average epoch time. We plot the results in Figure 7, where we can have following observations: 1) On Taobao and Koubei, GraphPL consistently outperforms FULL on all testing snapshots by a significant margin, which is a counter-intuitive finding. This confirms that on these datasets, taking time as an important factor and dynamically learning from temporal snapshots derive better recommendation accuracy. On Amazon, GraphPL and FULL generally perform on par. Specifically, GraphPL wins on the starting snapshots and for the rest, the performances are tightly close, which demonstrates that switching to the dynamic paradigm of GraphPL would yield potential performance improvements with minimal to no noticeable performance degradation, comparing to full training. 2) Our dynamic GraphPL framework greatly reduces the average epoch training time by a great margin, compared to FULL. Concretely the efficiency is boosted by 60x, 24x and 81x separately on Taobao, Koubei and Amazon. We attribute this advantage to the pre-training and prompt learning paradigm design. This advantage makes our model well-suited for efficient recommendation learning and prediction in real-world scenarios.

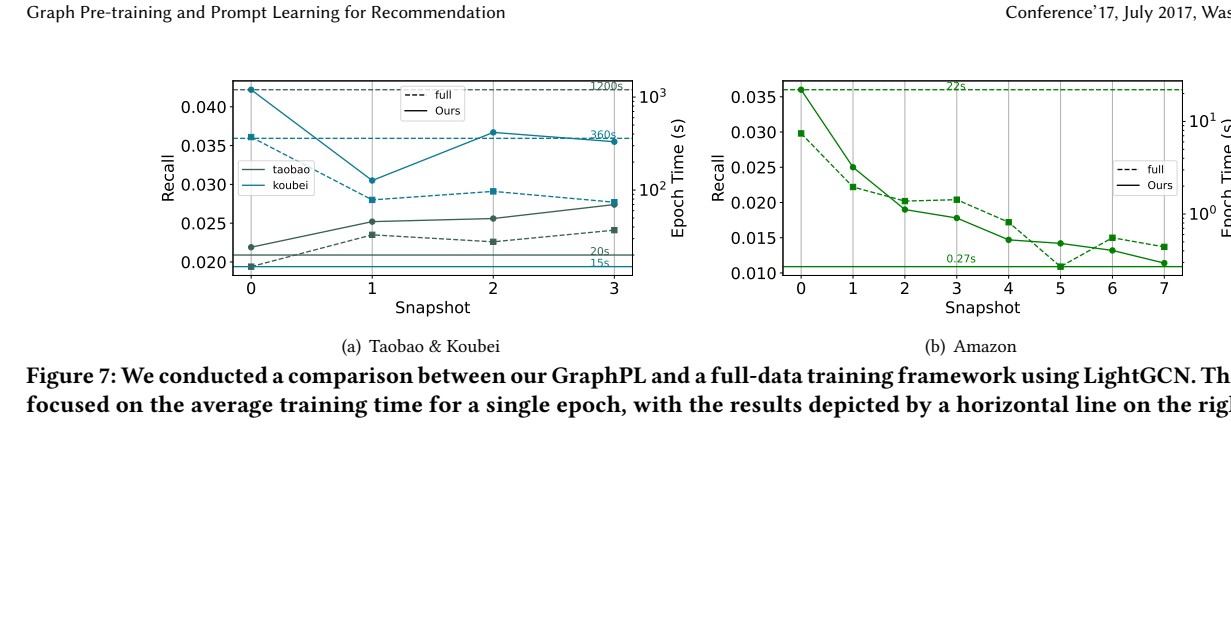

(a) Taobao & Koubei

(b) Amazon

Figure 7: We conducted a comparison between our GraphPL and a full-data training framework using LightGCN. The comparison focused on the average training time for a single epoch, with the results depicted by a horizontal line on the right Y-axis.

