# OpenReview forum: "Graph Pretraining and Prompt Learning for Recommendation"
_ACM.org/TheWebConf/2024/Conference — TheWebConf24_

### Official Review · Reviewer_k2rn · 2023-11-23

**Novelty:** 3
**Technical Quality:** 4

**Review:**

### Pros:
1. Code is open-source. Please kindly include all your re-implemented baselines in the codebase later.
2. The evaluation process is rigorous and convincing. Both online and offline experiments are conducted.
3. The idea of introducing positional encoding into the recommendation system is interesting.

### Cons:
1. Notations are not properly introduced. See questions and minors for more details.
2. The novelty of this paper is confusing. See question 3 for details. Even though the methodology itself is interesting, how the recsys task suits the methodology should be clearly stated.
3. The inference speed is the major drawback of the graph-based method in the real world. However, no efficiency metrics are reported during the inference stage.

### Minors
1. t_{u_3 v_2} in FIgure 2 is wrong, should be t_{u_3 v_1}.
2. Error in line 313. The notation is inconsistent with Eq 3.
3. How does the positional encoding help the proposed model's message aggregation?

**Questions:**

1. What is f in Eq 1?
2. Does node set V vary given different time slots? If yes, why is V in line 183, not V_1? If no, how do you explain the equation in line 210?
3. The reviewer is confused about where this paper should be categorized. Is this an improvement over temporal graph learning? Or is this an improvement over the dynamic recommendation method? If the former, then why did the author explicitly choose the recommendation task instead of the general graph learning task? If the latter, why does the author not compare with other non-graph-based methods on dynamic recommendation?

**Reviewer Confidence:**

3: The reviewer is confident but not certain that the evaluation is correct

**Scope:**

3: The work is somewhat relevant to the Web and to the track, and is of narrow interest to a sub-community

---

### Official Review · Reviewer_eiUQ · 2023-11-24

**Novelty:** 4
**Technical Quality:** 5

**Review:**

The article introduces GraphPL, a framework that combines graph pre-training and prompt learning for recommendation systems, which can incorporates parameter-efficient and dynamic graph pretraining with prompt learning. This framework addresses the challenge of evolving user preferences by incorporating a temporal prompt mechanism and a graph-structural prompt learning mechanism into the pre-trained graph neural network (GNN) model. The temporal prompt mechanism encodes time information to capture temporal context, while the graph-structural prompt learning mechanism enables the transfer of pre-trained knowledge to adapt to behavior dynamics without continuous incremental training.

Pros：

1.	The idea to incorporating a temporal prompt mechanism and a graph-structural prompt learning mechanism to capture temporal context and adapt to behavior dynamics is intuitive and straightforward.

2.	The method outperforms baseline methods and exhibits good performance in dynamics scenarios.

3.	GraphPL efficiently converges with the prompt learning paradigm, facilitating scalability and reducing the complexity of graph neural network models.

Cons:

1.	The main idea of this paper is to explore graph model pre-training methods in a dynamic recommendation environment. The dynamic scenario setting is very important, but it is not reflected in the title. It is suggested to include this scenario setting in the paper title.

2.	The performance improvement of GraphPL seems to vary across different backbone graph models, and the paper lacks an analysis of this phenomenon.

3.	In the paradigm of prompt learning, the length of the context is a crucial parameter. The authors only provided the settings for dataset division and lacked an analysis of the length of contextual information.

4.	The mathematical symbols appearing in lines 313~314 do not correspond to any formula in the article.

5.	How are the graph concatenation and sampling implemented in Equation (8)?

**Questions:**

See Cons.

**Ethics Review Description:**

No ethical issue.

**Reviewer Confidence:**

3: The reviewer is confident but not certain that the evaluation is correct

**Scope:**

4: The work is relevant to the Web and to the track, and is of broad interest to the community

---

### Official Review · Reviewer_prjL · 2023-11-24

**Novelty:** 5
**Technical Quality:** 5

**Review:**

**Summary:**

The paper introduces GraphPL, a framework designed to enhance Graph Neural Network (GNN)-based recommender systems. GraphPL integrates dynamic graph pre-training with prompt learning to address the limitations of current GNN recommenders in adapting to the evolving nature of user-item interactions. It includes a temporal prompt mechanism for encoding time in user-item interactions and a graph-structural prompt learning mechanism for transferring pre-trained knowledge without needing continuous incremental training. This approach aims to bridge the gap between offline and online recommendation scenarios, offering up-to-date and accurate recommendations in dynamic environments. The framework has been extensively tested, including a large-scale industrial deployment, demonstrating its effectiveness, robustness, and efficiency.

**Strength:**

1.	GraphPL's integration of dynamic graph pre-training and prompt learning addresses a crucial gap in adapting GNNs to evolving user preferences, which is a significant advancement.
2.	The framework's ability to encode temporal information and transfer pre-trained knowledge dynamically is a notable feature, enhancing its adaptability to user behavior changes.
3.	GraphPL's effectiveness in mimicking real-world scenarios and bridging the offline-online recommendation gap has been validated through extensive experimentation and industrial deployment.
4.	The lightweight and plug-in nature of GraphPL, coupled with its compatibility with various state-of-the-art recommenders, underscores its scalability and flexibility.

**Weakness:**
1.	The novel mechanisms introduced may add computational complexity to the baseline model's implementation. The computational costs associated with the pre-training and prompt learning processes are not adequately discussed. Understanding the trade-off between time cost and performance improvement is essential.
2.	Lack of interpretability about what kind of information may boost the improvement of performance with solid experiments.

**Questions:**

1. I wonder that while GraphPL is presented as a plug-in solution, whether the complexity involved in pretraining, integrating and fine-tuning it with existing systems could be challenging in industrial environment?
2. How does the system perform in a cold-start scenario.

**Reviewer Confidence:**

3: The reviewer is confident but not certain that the evaluation is correct

**Scope:**

4: The work is relevant to the Web and to the track, and is of broad interest to the community

---

### Official Review · Reviewer_FvcP · 2023-11-25

**Novelty:** 6
**Technical Quality:** 6

**Review:**

Summary: The novel technical contribution of this work is the proposal of GraphPL, a framework for recommendation systems that combines graph pre-training and prompt learning. GraphPL addresses the challenge of evolving user preferences and dynamic user-item interactions by incorporating temporal context and behavior dynamics into the pre-trained graph neural network (GNN) model. This work performs extensive and diverse set of experiments conducted to evaluate the GraphPL framework. The authors not only evaluate the framework on standard benchmark datasets but also perform a large-scale industrial deployment with A/B testing on an online platform.

Pros:
-	The introduction of the novel graph prompt learning paradigm in this work is an important and timely contribution to recommendation systems. By incorporating prompt learning mechanisms, the framework enables the recommendation system to adapt to changes in user preferences and behavior dynamics.

-	One positive aspect of this work is the proposal of GraphPL, a lightweight and parameter-efficient framework for recommendation systems. The authors emphasize its seamless integration into existing graph neural network (GNN) recommenders, making it easy to adopt without significant computational overhead or complex modifications.

-	Comprehensive experiments conducted and the real-world deployment in a large-scale industry recommendation environment. By covering various aspects, such as effectiveness, robustness, and efficiency, these experiments provide a comprehensive understanding of the framework's capabilities. The real-world deployment of the proposed framework in a large-scale industry recommendation environment is a significant positive aspect. Conducting A/B testing on an online platform further validates the effectiveness and performance of the framework in real-world scenarios.

-	Good quality of the write-up and well-motivated model design. By highlighting the challenges of existing recommender systems, the authors establish a good motivation for the development of GraphPL. Additionally, the release of the source code and model implementation, along with the detailed-described experimental setup and parameter settings, ensures the reproducibility of the results.

Cons:

-	The inclusion of dynamic GNN models as baselines by the authors is commendable. However, it would be advantageous to provide additional details on how to effectively integrate the dynamic graph encoder into the data modeling process, specifically in the context of bridging the pre-trained model with newly generated user behavior data. This information can include strategies for capturing temporal dependencies, and updating the graph representations to reflect the evolving user preferences over time.

In addition, it would be valuable to provide a more extensive discussion on the parameter sensitivity study in order to further elucidate the observation that GraphPL demonstrates greater sensitivity to specific hyper-parameters on the Amazon dataset compared to Taobao and Koubei.
-	To enhance the clarity and understanding of the methodology, it would be beneficial to include a notation table that provides a comprehensive list of key symbols used throughout the paper. Furthermore, to highlight the key steps involved in the dynamic learning framework of GraphPL, it would be advantageous to include a pseudocode section that presents a clear outline of the algorithmic steps involved in the dynamic graph encoder.

**Questions:**

-	The authors incorporate dynamic GNN models as baselines. However, to further enrich the paper, it would be advantageous to expand the discussion on effectively integrating the dynamic graph encoder into the data modeling process to accurately reflect the evolving user preferences with newly generated user behavior data.

-	Two suggestions can be implemented to enhance the methodology section. Firstly, including a notation table that provides a comprehensive list of key symbols used throughout the paper would improve clarity and ensure consistent interpretation. Secondly, incorporating a pseudocode section outlining the algorithmic steps of the dynamic graph encoder within the GraphPL's dynamic learning framework would effectively highlight the crucial steps involved.

**Reviewer Confidence:**

4: The reviewer is certain that the evaluation is correct and very familiar with the relevant literature

**Scope:**

4: The work is relevant to the Web and to the track, and is of broad interest to the community

---

### Official Review · Reviewer_SiEg · 2023-12-02

**Novelty:** 4
**Technical Quality:** 4

**Review:**

This paper addresses the dynamic modeling of time-evolving user preferences through graph-based methods.
The method is based on aggregating graph snapshots and propagating representations on the aggregated graphs to capture the temporal dynamics.
The final interaction prediction is calculated by the dot product between the user and item propagated representations.
The experiments are conducted not only on the three public datasets, but also on the A/B test.
The online A/B test also demonstrates the effectiveness of the proposed method over the online model.

The relevant studies are not comprehensive. This paper says "are not directly comparable to graph-based methods" and "existing sequential methods do not explicitly consider time in terms of daily or weekly intervals" in the section of related works. Actually, there are already some studies using GNNs for each time interval (e.g., snapshot) and connect these GNNs for sequential recommendation. However, this paper misses these important relevant studies. E.g.,
> Sequential Recommendation with Dual Side Neighbor-based Collaborative Relation Modeling, WSDM 2020.
> Learning Dual Dynamic Representations on Time-Sliced User-Item Interaction Graphs for Sequential Recommendation, CIKM 2021.

The comparisons are also not comprehensive. It is better to adopt some existing GNN-based recommendation methods that also consider time intervals.

Besides, it is hard to understand why the gating weights in Eq. 9 are random and non-learnable.

**Questions:**

Please see the above comments.

**Reviewer Confidence:**

4: The reviewer is certain that the evaluation is correct and very familiar with the relevant literature

**Scope:**

3: The work is somewhat relevant to the Web and to the track, and is of narrow interest to a sub-community

---

### Decision · Program_Chairs · 2024-01-22

**Decision:**

Accept

**Comment:**

This paper proposes a framework that combines graph pretraining and prompt learning. The idea is novel and interesting, and the evaluation is sufficient and well demonstrates that the framework has better performance in dynamics scenarios.